# The Enhanced Thermoelectric and Mechanical Performance of Polythiophene/Single-Walled Carbon Nanotube Composites with Polar Ethylene Glycol Branched-Chain Modifications

**DOI:** 10.3390/polym16070943

**Published:** 2024-03-29

**Authors:** Qing Yang, Shihong Chen, Dagang Wang, Yongfu Qiu, Zhongming Chen, Haixin Yang, Xiaogang Chen, Zijian Yin, Chengjun Pan

**Affiliations:** 1College of Chemistry and Chemical Engineering, Shenzhen University, Shenzhen 518060, China; 2110343048@email.szu.edu.cn (Q.Y.); chensher7@163.com (S.C.); wangdagang@szu.edu.cn (D.W.); 13720385426@163.com (H.Y.); 2110343112@email.szu.edu.cn (X.C.); 2110343060@email.szu.edu.cn (Z.Y.); 2School of Environment and Civil Engineering, Dongguan University of Technology, Dongguan 523808, China; qiuyf@dgut.edu.cn (Y.Q.); zmchen@dgut.edu.cn (Z.C.)

**Keywords:** composites, polythiophene, carbon nanotube, thermoelectric, polar side-chain

## Abstract

In order to develop flexible thermoelectric materials with thermoelectric and mechanical properties, in this study, we designed and synthesized polythiophene derivatives with branched ethylene glycol polar side-chains named P3MBTEMT, which were used in combination with single-walled carbon nanotubes (SWCNTs) to prepare composite thin films and flexible thermoelectric devices. A comparison was made with a polymer named P3(TEG)T, which has a polar alkoxy linear chain. The UV-vis results indicated that the larger steric hindrances of the branched ethylene glycol side-chain in P3MBTEMT could inhibit its self-aggregation and had a stronger interaction with the SWCNTs compared to that of P3(TEG)T, which was also confirmed using Raman spectroscopy. When the mass ratio of SWCNTs to P3MBTEMT was 9:1 (represented as P3MBTEMT/SWCNTs-0.9), the composite film exhibited the highest thermoelectric properties with a power factor of 446.98 μW m^−1^ K^−2^, which was more than two times higher than that of P3(TEG)T/SWCNTs-0.9 (215.08 μW m^−1^ K^−2^). The output power of the thermoelectric device with P3MBTEMT/SWCNTs-0.9 was 2483.92 nW at 50 K, which was 1.66 times higher than that of P3(TEG)T/SWCNTs-0.9 (1492.65 nW). Furthermore, the P3MBTEMT/SWCNTs-0.5 showed superior mechanical properties compared to P3(TEG)T/SWCNTs-0.5. These results indicated that the mechanical and thermoelectric performances of polymer/SWCNT composites could be significantly improved by adding polar branched side-chains to conjugated polymers. This study provided a new strategy for creating high-performing novel flexible thermoelectric materials.

## 1. Introduction

Thermoelectric (TE) materials, a sort of green energy material that has the ability to convert “thermal energy” into “electrical energy” without the need for external power, have received a significant attention in the field of materials science [1,2,3]. Flexible and wearable thermoelectric devices represent a hot research area within the realm of thermoelectric materials [4,5,6,7]. These devices can generate a voltage difference from the variation in temperature between the device and the environment alone, thereby achieving signal transmission without the need for an external power source. Moreover, they can operate normally on heat source surfaces with complex curvature changes. As such, flexible thermoelectric devices are required to have both an excellent TE performance and outstanding mechanical properties [7,8]. Due to their inherent flexibility and low thermal conductivity, polymeric materials are suited for application in the manufacturing of these flexible TE devices. [9]. Their TE performance is usually assessed by the power factor PF (PF = *S*^2^*σ*, where *S* = Seebeck coefficient and *σ* = conductivity). However, polymers tend to have a low conductivity, and one strategy for enhancing their thermoelectric performance is to combine them with high-conductivity materials such as single-walled carbon nanotubes (SWCNTs) [10].

With their exceptional conductivity, stable Seebeck coefficient, and good flexibility, SWCNTs have become a star material in TE research. However, their high thermal conductivity is a problem that needs to be urgently resolved [11]. In order to create organic/inorganic composite TE materials with a high conductivity, a large Seebeck coefficient, and a relatively low thermal conductivity, SWCNTs are frequently utilized as an inorganic filler in combination with low thermal conductive polymers in the field of thermoelectric materials. A win–win situation is thereby achieved [12]. Research on conjugated polymer/SWCNT composite materials has been widely reported [13], with the most common examples being poly(3-hexylthiophene) (P3HT) [14,15], polypyrrole (PPy) [16,17], poly(3,4-ethylenedioxythiophene):poly(styrenesulfonate) (PEDOT:PSS) [18,19], and polyaniline (PANI) [20,21]. The π-π interactions between conjugated polymers and SWCNTs can facilitate the interface interaction between nanoparticles and polymers, greatly enhancing their carrier mobility [22].

When designing conjugated polymers with a high TE performance, the role of side-chains is as important as that of the conjugated backbone. In the past few years, a growing number of reports have suggested that the introduction of polar side-chains can enhance the TE performance of polymer/SWCNT composites to varying degrees [23,24,25]. Hao et al. synthesized a polythiophene derivative with polar alkoxy linear chains, named PMEET. Compared to P3HT, they found that PMEET could interact more strongly with SWCNTs, with the power factor of PMEET/SWCNTs (121 μW m^−^^1^ K^−^^2^) being twice as large as that of P3HT/SWCNTs (65 μW m^−^^1^ K^−^^2^) [23]. By introducing alkoxy linear chains or macrocyclic side-chains into benzodithiophene (BDT), Wu et al. found that the latter could inhibit the self-assembly of polymers, promote the π-π interactions between polymers and SWCNTs, and enhance the TE performance (137.7 μW m^−^^1^ K^−^^2^) by 1.8 times [25]. Furthermore, the branching of the side-chains could aid in plasticizing and softening [26], which are of great significance for the development of flexible materials. Currently, in the field of organic TE materials, studies on branched polar side-chains mainly involve chemical doping [27,28]. The introduction of side-chains can enhance the TE performance by increasing the compatibility between polymers and dopants. However, studies on the influence of polar branched side-chains on the thermoelectric and mechanical properties of polymer/SWCNT composites have not yet been reported.

In the present study, we designed and synthesized two polythiophene derivatives with similar number average molecular weights (*M*_n_) using side-chain engineering: poly(3-(2-(2-(2-methoxyethoxy)ethoxy)ethoxy)methylthiophene) (P3(TEG)T) (containing polar alkoxy linear chains) and poly(3-(1,3-bis(triethoxymethoxy)propan-2-yloxy)methylthiophene) (P3MBTEMT) (containing polar ethylene glycol branches). We combined these derivatives with SWCNTs to prepare TE thin film materials and flexible TE-based devices. At the optimal composite mass ratio of P3MBTEMT to SWCNTs, which was 1:9, the power factor (PF) of the P3MBTEMT/SWCNT composite thin film was 446.98 μW m^−^^1^ K^−^^2^, which was 2.07 times larger than that of P3(TEG)T/SWCNTs (215.08 μW m^−^^1^ K^−^^2^) and superior to the pure SWCNTs (344.97 μW m^−^^1^ K^−^^2^). The TE-based device P3MBTEMT/SWCNTs had a maximum output power of 2481.97 nW at a temperature difference (ΔT) of 50K, which was 1.66 times larger than that of P3(TEG)T/SWCNTs (1492.65 nW). The mechanical performance of P3MBTEMT/SWCNTs was superior to that of P3(TEG)T/SWCNTs for a composite mass ratio of 1:1. This demonstrated that it is feasible to use side-chain engineering and the introduction of polar branched structures to enhance the thermoelectric and mechanical properties of conjugated polymers.

## 2. Experimental Section

### 2.1. Polymer/SWCNT Composite Film Preparation

The SWCNTs were dissolved in chlorobenzene (1 mg/mL) and, thereafter, were treated ultrasonically for 5 h at room temperature (the water in the ultrasonic cleaning machine was changed every hour). The solution was then stirred overnight to obtain a uniformly dispersed SWCNT solution. Glass bottles were used to prepare the composite solutions with the polymer. The polymer-to-SWCNTs mass ratios were 3:7, 5:5, 7:3, and 9:1. The preparation of the solutions was followed by an additional 2 h of ultrasonical treatment to ensure the uniform mixing of the polymer and the SWCNTs. A pipette was used to add 120 uL of the solution to a 1 cm × 1 cm glass slide, which was left to dry naturally at room temperature. This resulted in a composite film with a thickness of approximately 2 μm. In this study, the polymer/SWCNT composite films were labeled according to the mass percentage of the SWCNTs. For example, when the mass ratio of P3(TEG)T-to-SWCNTs was 1:9, it was labeled as P3(TEG)T/SWCNTs-0.9.

### 2.2. Fabrication of p-Type Polymer/SWCNTs TE Devices

The flexible polyimide (PI) film was cut into 1 cm × 4 cm strips. A pipette was, thereafter, used to spread 300 µL of the well-dispersed polymer/SWCNTs composite solution on the PI film, which was left to dry at room temperature. Ten 1 cm × 4 cm composite films were then attached to a 8 cm × 24 cm PI film surface in sequence with double-sided tape, leaving a 1 cm spacing between each pair. Subsequently, copper foil tape was used to join the films in series, and conductive silver glue was used to ensure a solid connection at the point where the copper foil tape and the composite film met. The fabricated thermoelectric film device is shown in Appendix A.

## 3. Results and Discussion

### 3.1. Synthesis and Characterization of Polymers

Two polymers with similar molecular weights (*M*_n_), P3(TEG)T and P3MBTEMT (Figure 1a), were prepared using a simple Grignard metathesis polymerization (GMIR) method. The *M*_n_ of the polymers, P3(TEG)T and P3MBTEMT, were determined as 18.4 kDa and 18.9 kDa, respectively, using gel permeation chromatography (GPC). Their polydispersity indices (PDI) were 1.53 and 1.84, respectively (Appendix A). Furthermore, the structures of the polymers were characterized using proton nuclear magnetic resonance spectroscopy (^1^H NMR) (Appendix A). The combination of the ^1^H NMR and GPC results validated that the synthesized polymers met the research requirements. A thermogravimetric analysis (TGA) demonstrated that the decomposition temperatures of P3(TEG)T and P3MBTEMT (defined as the temperature at which the sample’s weight dropped to 95% of its original weight) were 247.74 °C and 255.20 °C, respectively (Appendix A), which indicated that the polymers had a good thermal stability. Figure 1b shows the normalized ultraviolet-visible (UV-vis) spectra of P3(TEG)T and P3MBTEMT. The strongest absorption peaks of P3(TEG)T and P3MBTEMT in the solvent-free state were at 431 nm and 447 nm, respectively. Both of these peaks resulted from π-π* transitions in the polythiophene main-chain backbone. There was no significant change in P3MBTEMT in either the tetrahydrofuran (THF) solution or the solvent-free state, which was due to the large branched alkoxyl side-chains that were wrapped around the thiophene backbone. The significant sterical hindrances could inhibit the self-assembly of P3MBTEMT, which was beneficial for the homogeneous blending with SWCNTs.

### 3.2. Raman Spectroscopy

Figure 2 shows the Raman spectra of the polymer/SWCNT composite films with various SWCNT contents. The symmetric stretching vibration peak of the C_α_ = C_β_ double bond of the thiophene group in the polymer backbone was the absorption peak at around 1450 cm^−1^ [29]. Furthermore, the stretching vibration peak of C-O-C on the side-chain coincided with the absorption peak at around 1185 cm^−1^. The strong absorption peak at around 1590 cm^−1^ and weak absorption peak at around 1570 cm^−1^ constituted the typical G band of the semiconducting SWCNTs (i.e., G^+^ and G^−^, respectively). They were formed by thesp^2^ hybridized carbon atoms vibrating within the SWCNTs’ hexagonal lattice [30,31]. Thus, there was a significant amount of carbon atoms that were sp^2^ hybridized, which facilitated the charge transport. There was no noticeable D band at around 1350 cm^−1^ for the pure SWCNTs, which indicated that the SWCNTs used here were of a high quality. They were free from defects due to the carbonized tube walls and particles. Similarly, there were no obvious defects in the polymer/SWCNT composite films. This indicated that no significant structural defects were formed during the mixing procedure, and that the components were fully blended [32,33], which facilitated the charge transport and improved the thermoelectric performance. Furthermore, the position of the G band for P3MBTEMT/SWCNTs-0.5 (1589.29 cm^−1^) showed a blue shift in relation to the pure SWCNTs (1591.28 cm^−1^). However, there was no change for P3(TEG)T (1591.28 cm^−1^), which suggested a stronger interaction between P3MBTEMT and the SWCNTs. As can be seen in Figure 2b, the content of the SWCNTs increased in the P3MBTEMT/SWCNT composite films, the symmetric stretching vibration peak of the C_α_ = C_β_ double bond had a slight red shift, and the G band peak intensity increased and showed a slight blue shift relative to the SWCNTs. This could be attributed to the impact of the π-π interactions in the interface formed by P3MBTEMT and the SWCNTs in the composite films.

### 3.3. X-ray Diffraction

The X-ray diffraction (XRD) spectra of the polymer/SWCNT composite films with different SWCNT contents are shown in Figure 3. P3(TEG)T and P3MBTEMT only had a diffuse peak at 25°, which indicated that they were amorphous polymers without a long-range ordered structure. This was reasonable, since P3MBTEMT had larger alkyl–oxygen side-chains. This side-chain structure would most probably have affected the interactions of the P3MBTEMT backbone, preventing the formation of a closely packed structure. Furthermore, the SWCNTs showed a typical weak diffraction peak at 26.7°, which most probably originated from the catalyst added in the preparation of the SWCNTs [34]. For the P3MBTEMT/SWCNT composite, new peaks appeared at 12.7°, 17.6°, 19.6°, 20.7°, 21.4°, 23.6°, and 26.5°. The diffraction peak of the SWCNTs moved to 26.5°, which indicated that the strong interfacial interaction between the P3MBTEMT polymer and the SWCNTs in the composite material allowed for P3MBTEMT to be evenly coated on the SWCNT bundle. An ordered structure was thereby formed, which was beneficial for the charge transfer at the interface.

### 3.4. Microscopic Morphology Studies

Figure 4 shows the surface morphologies of the polymer/SWCNT composite films with different SWCNT contents. The surfaces of the P3(TEG)T and P3MBTEMT films exhibited an aggregation-like morphology (Figure 4a,b). On the other hand, the SWCNT sample showed a uniform distribution without any aggregation (Figure 4g), indicating a good dispersion of SWCNTs in the source solution. As shown in Figure 4c–f, with an increase in the SWCNT content, the P3MBTEMT/SWCNTs film became uniformly distributed, without any aggregation on its surface [35]. The composite film exhibited a network structure of SWCNTs, with apparent fibrous structures. This was due to the wrapping of the polymer around the SWCNT bundles, causing the diameter of the SWCNT bundles to increase and the connections between the bundles to become more numerous. Thus, neighboring SWCNT bundles became closely connected via strong π-π interactions, thereby forming a conductive network that was favorable for charge transmission. This could effectively enhance the conductivity of the composite material, thereby improving the thermoelectric performance of the composite film.

### 3.5. Mechanical Properties

To compare the mechanical properties of the composite films, they were subjected to mechanical tensile testing and bending testing. As shown in Figure 5a,b, the composite films exhibited superior mechanical tensile properties compared to the pure carbon tubes. This was due to the addition of the polymers, P3MBTEMT and P3(TEG)T, resulting in strong π-π interfacial interactions between the SWCNT bundles and the polymers, thus enhancing the effective stress of the composite films [36]. In addition, P3MBTEMT/SWCNTs showed an improved tensile modulus, breakage stress, and breakage elongation, as compared to P3(TEG)T/SWCNTs-0.5. The maximum tensile modulus reached as high as 6.11 MPa. This large value could most probably be explained by the larger alkoxy side-chain structure of P3MBTEMT, which could be altered during the tensile process to provide internal friction in the composite film. These results demonstrated the superior mechanical properties of the P3MBTEMT/SWCNTs-0.5 composite film. Furthermore, Figure 5c,d show that the rate of change of the composite film resistance was small under bending for different bending times and for different bending curvatures (i.e., radii), respectively. These findings showed the excellent flexibility and durability of the P3MBTEMT/SWCNTs-0.5 and P3(TEG)T/SWCNTs-0.5 composite films.

### 3.6. Thermoelectric Properties

The TE properties of the P3(TEG)T/SWCNT and P3MBTEMT/SWCNT composite films at room temperature were found (Figure 6a,b). The composite films’ Seebeck coefficients were all larger than 0, indicating that they were p-type TE materials with conducting hole carriers. The electrical conductivity of the composite films increased with an increased content of SWCNTs. The reason for this was that the addition of SWCNT bundles provided more channels for rapid electron transfer. Furthermore, the Seebeck coefficient of P3MBTEMT/SWCNTs was generally larger than that of P3(TEG)T/SWCNTs. Notably, P3MBTEMT/SWCNTs-0.9 had a higher conductivity than the pure SWCNTs. This could be explained by the branched ethylene glycol polar side-chains of P3MBTEMT, which could not only promote electron transfer, but also provide a better solution processability, therefore fostering strong π-π interactions between P3MBTEMT and the SWCNTs at the interfaces. Moreover, P3MBTEMT/SWCNTs had a slightly lower Seebeck coefficient than the SWCNTs, which was most probably caused by the strong binding between the wrapped P3MBTEMT and the SWCNT bundle. This strong interaction helped to achieve a lower contact resistance [37], which, in turn, might have led to an increase in the charge concentration, thereby causing a slight decrease in the Seebeck coefficient [38]. Finally, the calculated PF value of P3MBTEMT/SWCNTs-0.9 was 446.98 μW m^−1^ K^−2^, which was higher than that of the pure SWCNTs (344.97 μW m^−1^ K^−2^). The flexible p-type TE devices were fabricated on a PI flexible film substrate with P3(TEG)T/SWCNTs-0.9 and P3MBTEMT/SWCNTs-0.9, which had the highest power factor. As can be seen in Figure 6c,d, for a temperature difference (ΔT) of 50 K, the device reached its maximum output power and open-circuit voltage. The maximum power of P3MBTEMT/SWCNTs-0.9 was 2481.97 nW, which was 1.66 times larger than that of P3(TEG)T/SWCNTs-0.9 (1492.65 nW). Furthermore, the maximum open-circuit voltage was 25.35 mV, which was higher than that for the device P3(TEG)T/SWCNTs-0.9 (20.73 nV). Finally, a simple TE device consisting of three TE elements was fabricated (Appendix A). A change in human body temperature was simulated by finger pressure. As different numbers of fingers were in contact with the device surface, temperature differences were obtained at both ends of the device (Appendix A). Consequently, there was also a change in the voltage at both ends of the device (Figure 6e).

## 4. Conclusions

This work synthesized P3TEGT (containing polar alkoxy linear chains) and P3MBTEMT (containing polar ethylene glycol branches) and combined them with SWCNTs to prepare flexible TE materials. The branched ethylene glycol side-chain enhanced the π-π interactions between the polymer and SWCNTs, achieving a higher electrical conductivity than that of pure SWCNTs. The PF value of P3MBTEMT/SWCNTs-0.9 reached 446.98 μW m^−1^ K^−2^, which was higher than that of P3(TEG)T/SWCNTs-0.9 (215.08 μW m^−1^ K^−2^) and the pure SWCNTs (344.97 μW m^−1^ K^−2^). The larger alkoxy side-chain did not only promote electron transmission, but also provided a larger flexibility to the polymer. P3MBTEMT/SWCNTs-0.5 had a higher tensile modulus than P3(TEG)T/SWCNTs-0.5, with a value of 6.11 MPa. In summary, the addition of polar branched side-chains to conjugated polymers could improve the thermoelectric and mechanical properties of composite films. This type of side-chain engineering has provided a strategy for the development of new flexible thermoelectric materials.

## Figures and Tables

**Figure 1 polymers-16-00943-f001:**
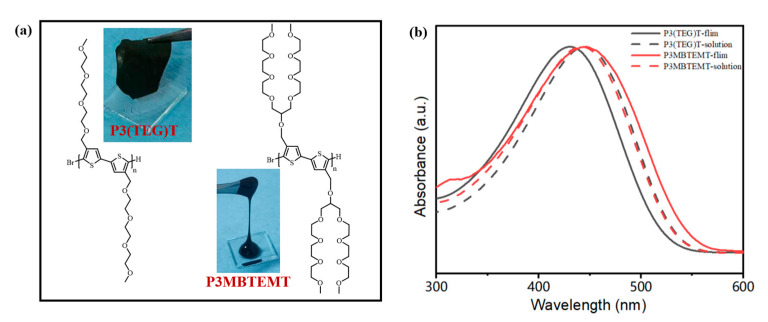
(**a**) Chemical structures and ambient states of P3(TEG)T and P3MBTEMT. (**b**) Normalized UV–vis absorption spectra of P3(TEG)T and P3MBTEMT in the THF solution and in the thin film states.

**Figure 2 polymers-16-00943-f002:**
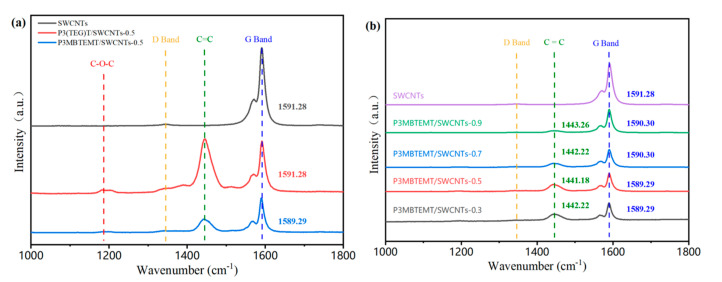
(**a**) Raman spectra of SWCNTs, P3(TEG)T/SWCNTs-0.5, and P3MBTEMT/SWCNTs-0.5. (**b**) Raman spectra of SWCNTs, P3MBTEMT/SWCNTs-0.9, P3MBTEMT/SWCNTs-0.7, P3MBTEMT/SWCNTs-0.5, and P3MBTEMT/SWCNTs-0.3.

**Figure 3 polymers-16-00943-f003:**
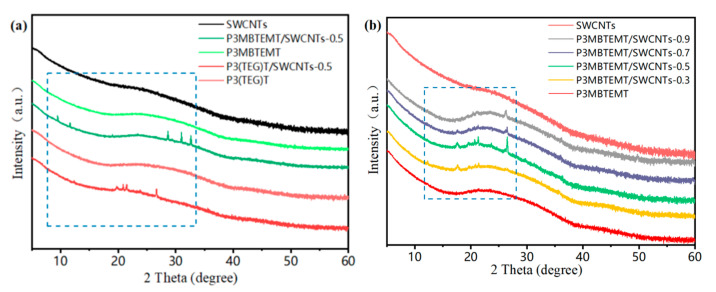
XRD patterns of (**a**) SWCNTs, pristine P3(TEG)T, pristine P3MBTEMT, P3(TEG)T/SWCNTs-0.5, and P3MBTEMT/SWCNTs-0.5. XRD patterns of (**b**) SWCNTs, pristine P3MBTEMT, and P3MBTEMT/SWCNT composite films.

**Figure 4 polymers-16-00943-f004:**
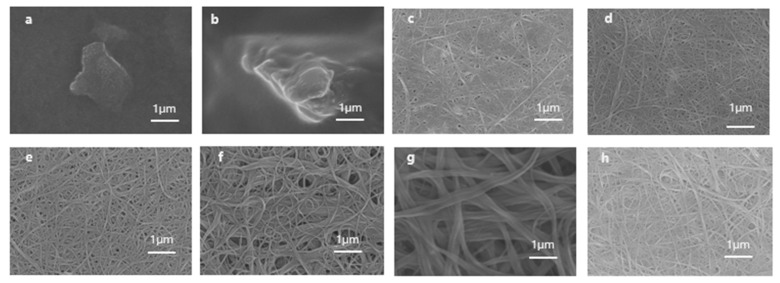
SEM images of surface morphologies for (**a**) pristine P3(TEG)T, (**b**) pristine P3MBTEMT, (**c**) P3MBTEMT/SWCNTs-0.3, (**d**) P3MBTEMT/SWCNTs-0.5, (**e**) P3MBTEMT/SWCNTs-0.7, (**f**) P3MBTEMT/SWCNTs-0.9, (**g**) SWCNTs, and (**h**) P3(TEG)T/SWCNTs-0.5.

**Figure 5 polymers-16-00943-f005:**
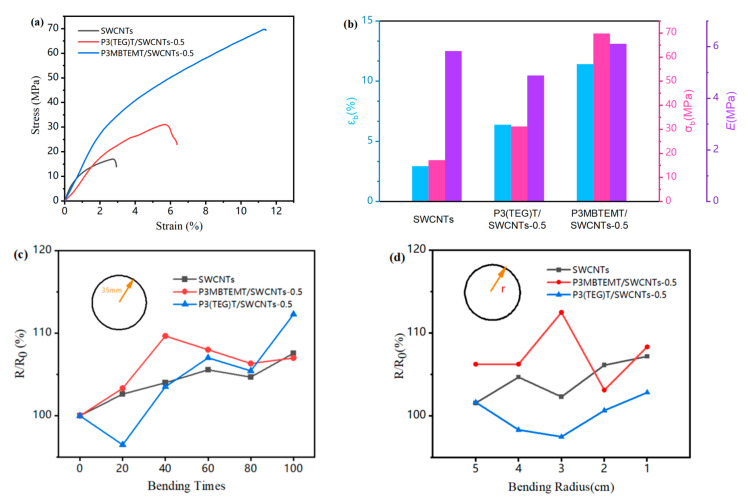
For SWCNTs, P3(TEG)T/SWCNTs-0.5, and P3MBTEMT/SWCNTs-0.5; (**a**) stress–strain curves. (**b**) Tensile modulus, breakage stress, and breakage strain. (**c**) Resistance variation with bending times. (**d**) Resistance variation at different bending radii.

**Figure 6 polymers-16-00943-f006:**
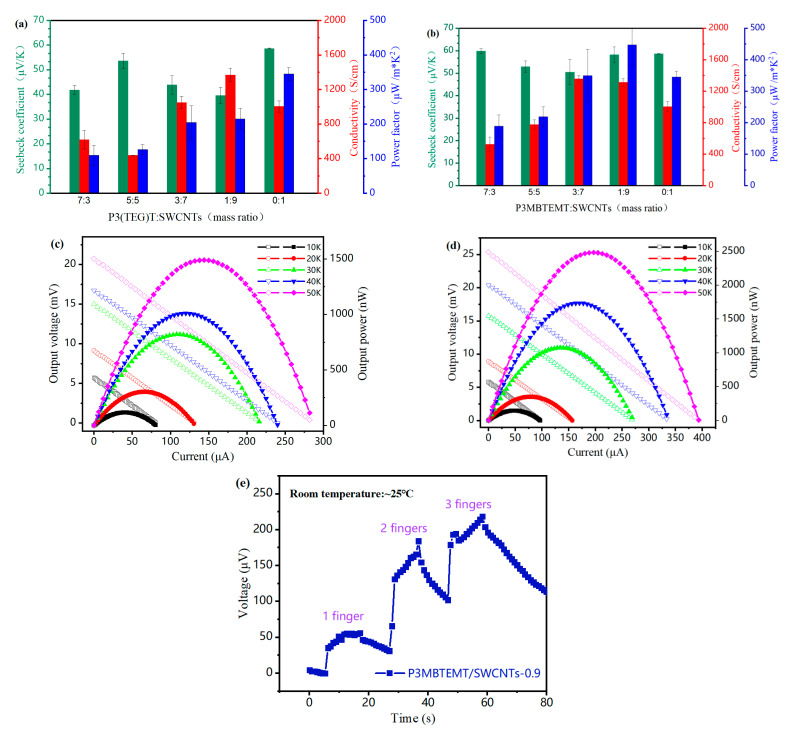
*σ*, *S*, and PF values of (**a**) P3(TEG)T/SWCNT composite films and (**b**) P3MBTEMT/SWCNT composite films with various SWCNT mass ratios. Voltage–current (open shapes) and power density–current (solid shapes) output curves of (**c**) P3(TEG)T/SWCNTs-0.9 device and (**d**) P3MBTEMT/SWCNTs-0.9 device at different temperatures. (**e**) Voltage–time graph of a simplified P3MBTEMT/SWCNTs-0.9 device after being touched with fingers.

## Data Availability

The data presented in this study are available on request from the corresponding author.

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
