# Peer review of "The Enhanced Thermoelectric and Mechanical Performance of Polythiophene/Single-Walled Carbon Nanotube Composites with Polar Ethylene Glycol Branched-Chain Modifications"

_polymers, 2024, doi:10.3390/polym16070943_

Round 1
Reviewer 1 Report
The article entitled "Enhanced Thermoelectric and Mechanical Performance of Poly-2 thiophene/Single-Walled Carbon Nanotube Composites by Po- 3 Lar Ethylene Glycol Branched Chain Modifications" concerns the current its high thermoelectric and mechanical properties. Therefore, the choice of topic does not raise objections. The Introduction section is quite extensive and well written, with links to other papers with a clear indication of the results achieved. Experimental section is described correctly. However, the authors used a drop casting technique to produce the layers. This technique is of course acceptable, however, for more advanced research, the authors should consider the production of layers using the doctor blade method or multi-stage spin coating for better uniformity of layer thickness. The Results and Discussion section is comprehensively described, although the subsections "Microscopic morphology studies", "Mechanical properties" and "Thermoelectric properties" are somewhat missing references to other papers as discussions. The "Conclusion" section does not raise any major objections, it clearly describes what has been achieved. The literature is relatively modern. There are several editorial errors. To sum up, in my opinion, after editorial checking and adding a some references to discusion , the text is suitable for publication.
There are several editorial and language errors that do not directly affect the substantive value of the work.
Reviewer 2 Report
The paper reported novel P3MBTEMT/SWCNTs thermoelectric materials for flexible and high-output thermoelectric devices. There are some mistypes, but manuscript is well-written and informative to develop thermoelectric polymers. Authors also fabricated flexible devices and showed the increased tensile strength, elongation and bending properties. However, there need more explanation to understand the thermoelectric behaviors of P3MBTEMT/SWCNTs.
1. The absorption of P3(TEG)T (Figure 1b) was blue-shifted in the film states. Please state the reasonable explanation.
2. How about the HOMO and LUMO energy levels of the P3(TEG)T and P3MBTEMT? Can you measure the energy level from cyclic voltammetry?
3. In Figure 3a-b, authors mentioned that P3(TEG)T is a crystalline polymer, but reflection pattern is not regular at all. Are you sure that it comes from the P3(TEG)T? Please state reasonable explanation.
4. Interestingly, both tensile strength and ductility were improved in the P3MBTEMT/SWCNTs. Why did P3(TEG)T/SWCNTs based-on crystalline polymer show weaker tensile strength?
5. In general, there is trade-off relationship between Seebeck coefficient and conductivity. That means that higher ratio of SWCNT should show higher electrical conductivity, but lower seebeck coefficient. However, in Figure 6a-b, there are no any close relationship among electrical conductivity, seebeck coefficient and the blending ratio of polymer/SWCNTs. Why are both high electrical conductivity and seebeck coefficient obtained at 1:9 ratio of P3MBTEMT/SWCNTs? In addition, the PF of P3(TEG)T/SWCNTs was lower than that of 100% SWCNTs. Why?
Minor correction is required.
Reviewer 3 Report
The Authors prepared polythiophene derivatives with branched ethylene glycol polar side-chains and used them in combination with SWCNT fillers for the fabrication of flexible thermoelectric composites. The results showed that the performance of such thermoelectric composites may be significantly improved by adding polar branched side-chains to conjugated polymers. This work is well-written, shows interesting and reliable results, and will be interesting for the researchers dealing with the development of flexible thermoelectric composites. I recommend to publish this work as it is.
